# Emergence of Spatial Representation in an Actor-Critic Agent with Hippocampus-Inspired Sequence Generator

**Xiao-Xiong Lin**[*]  **Yuk-Hoi Yiu**  **Christian Leibold**

Faculty of Biology & Bernstein Center Freiburg & BrainLinks-BrainTools Center
University of Freiburg, Freiburg, Germany

## Abstract

Sequential firing of hippocampal place cells is often attributed to sequential sensory drive along a trajectory, and has also been attributed to planning and other cognitive functions. Here, we propose a mechanistic and parsimonious interpretation to complement these ideas: hippocampal sequences arise from intrinsic recurrent circuitry that propagates transient input over long horizons, acting as a temporal memory buffer that is especially useful when reliable sensory evidence is sparse. We implement this idea with a minimal sequence generator inspired by neurobiology and pair it with an actor–critic learner for egocentric visual navigation. Our agent reliably solves a continuous maze without explicit geometric cues, with performance depending on the length of the recurrent sequence. Crucially, the model outperforms LSTM cores under sparse input conditions (16 channels, $\sim 2.5\%$ activity), but not under dense input, revealing a strong interaction between representational sparsity and memory architecture. Through learning, units develop localized place fields, distance-dependent spatial kernels, and task-dependent remapping, while inputs to the sequence generator orthogonalize and spatial information increases across layers. These phenomena align with neurobiological data and are causal to performance. Together, our results show that sparse input synergizes with sequence-generating dynamics, providing both a mechanistic account of place cell sequences in the mammalian hippocampus and a simple inductive bias for reinforcement learning based on sparse egocentric inputs in navigation tasks. Code: `https://github.com/xiaoxionglin/SF_hipposlam`

## 1 Introduction

Hippocampal place cells track the animal's location during navigation (O'Keefe & Dostrovsky, 1971) and they fire in sequence reflecting the behavioral order of the place fields (Foster & Wilson, 2007). Spatial locations are thereby thought to serve as anchors for episodic memories (Aronowitz & Nadel, 2023), a view reinforced by observations of "look-ahead" sequence replay linked to trajectory planning (Foster & Wilson, 2007; Kay et al., 2020). On the other hand, hippocampal neurons are also found to fire at successive moments not explainable by location alone (Eichenbaum, 2014), suggesting the place cell sequences could reflect timing rather than spatial input.

The intertwined spatial and temporal representations in hippocampus has been touched upon in many recent computational models. Successor representations interpret place cells as predictive states (Stachenfeld et al., 2017; Mattar & Daw, 2018); reservoir models emphasize pre-existing dynamics in shaping place cell sequences (Leibold, 2020); probabilistic approaches model place cells as latent states inferred from successive inputs (Raju et al., 2024); and self-supervised methods refine spatial tuning by exploiting temporal smoothness of the trajectory (Wang et al., 2024). While these approaches reproduce place-like activity and even sequential patterns, they rarely address explicitly where hippocampal sequences originate. Many bottom-up studies instead emphasize that structured representations emerge from circuit dynamics and physiological constraints, suggesting mechanistic

---

[*]Correspondence to: `ln.xxng@gmail.com`

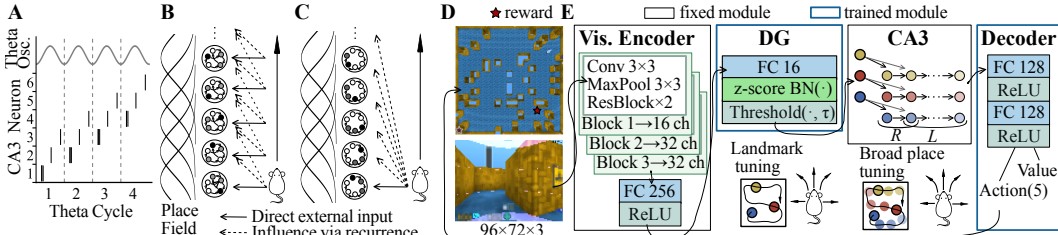

Figure 1: Model summary. **A**: Illustration of theta sequences observed in rodent hippocampus. In each theta cycle, $R = 3$ neurons are activated and the activation propagates over $L = 4$ theta cycles in a sequence of $\ell = L + R - 1 = 6$ neurons. **B**: The theta sequences are thought to be driven by sequential inputs despite the recurrent connections in hippocampus. **C**: The recurrent connections could support generating long horizon sequential activity without sequential external inputs. **D**: A virtual environment ($19 \times 19$ tiles) was constructed using DeepMind Lab with walls randomly placed on 15 % of the tiles. Wall layouts are kept fixed across episodes, with an invisible reward near the bottom right. **E**: The agent receives a first person perspective visual input that is processed via an encoder (shallow ResNet with 3 convolutional blocks; matching the SOTA in DeepMind Lab environment (Espeholt et al., 2018), pretrained and fixed in our experiments). These outputs are linearly mapped to $F = 16$ features (FC: fully connected layer), and then sparsified using batch normalization and high thresholding ($\tau = 2.43$), such that the percentage of activation ($\sim 2.5\%$) matches the sparse activity of DG granular cells that project to CA3. CA3 is modeled via sequences of neuron activations, each sequence evoked by a distinct DG input feature. The activity of all CA3 neurons are then flattened and linearly mapped to the decoder multilayer perceptron. CA3 is hard coded to isolate the effect of long range integration. The DG and decoder modules are trained.

models are needed to link anatomy, dynamics, and representational patterns (Lin et al., 2023; Buzsáki & Tingley, 2018; Schaeffer et al., 2022).

We propose a parsimonious account directly addressing this question: hippocampal sequences arise from intrinsic recurrent circuitry in CA3 that can propagate activity over long timescales in the absence of input (Fig. 1A-C). The CA3 sequence generator receives sparse inputs from dentate gyrus (DG) to yield localized spatial codes in CA3 that support navigation (Leibold, 2020; 2022). The sparse-input regime is not incidental but central: it reflects the ecological reality that navigation is often guided by only a few reliable landmarks amid abundant sensory noise; the biology of DG granule cells, which fire at extremely low rates; and the computational advantages of high-capacity, low-interference codes that promote compositionality and generalization.

This mechanism mirrors recent key ideas from machine learning. State-space models and structured linear RNNs preserve long-range information by expanding inputs into a high-dimensional temporal feature space before compressing them via shallow nonlinear readouts (Fu et al., 2022; Gu et al., 2020; 2021). Our model resonates with this principle: DG sparsification provides a low-activity code that is sustained and expanded by intrinsic recurrence, offering a rich set of features for downstream policy learning.

To test this hypothesis, we implement an agent with a DG-like sparsification module, a recurrent sequence generator (CA3 proxy), and an actor–critic learner for egocentric navigation. We show that sequence generation and sparse input synergize, outperforming LSTMs of comparable size in the sparse-input regime, while LSTMs remain competitive under dense input. Moreover, place fields, DG orthogonalization, and task-dependent remapping emerge naturally during training.

These results suggest that a sequence-based reservoir, inspired by CA3, is well suited for constructing spatial representations from sparse low-bandwidth inputs. The synergy between sparse coding and intrinsic sequence dynamics thus offers both a mechanistic explanation for hippocampal sequences and a simple inductive bias for reinforcement learning in navigation tasks.

## 2 METHODS

### 2.1 VIRTUAL ENVIRONMENT

We consider an agent navigating a continuous environment with sparse obstacles and uniform visual textures. The environment was simulated using DeepMind Lab (Beattie et al., 2016) (Figure 1D). The wall layouts were kept fixed for repeated episodes unless otherwise stated. The environment was designed such that spatial relations between locations cannot be trivially inferred from the similarity of their corresponding visual features and that it allows multiple possible trajectories from a given location to the target.

In each episode, the agent was initially placed at a random location at least 5 units from the goal. An episode is finished when the agent reaches the goal or a maximum of 7200 frames (with action repeat=8 corresponding to a maximum of 900 action steps, i.e. time steps in the model). For the map displayed in Fig. 1D, the reward was placed near the bottom right corner and then moved to the lower left corner to test generalization.

### 2.2 VISUAL PROCESSING

The visual encoder is intended to mimic visual cortical preprocessing, extracting general-purpose visual features for the hippocampus. We pretrained a ResNet encoder (He et al., 2015) in combination with an LSTM core as in Espeholt et al. (2018) and kept it fixed when training our hippocampus model. We also tested a variant using the layer 2 output from ResNet pretrained on Imagenet. The model produces similar performance (Fig. A2).

### 2.3 HIPPOCAMPUS MODEL

**Dentate gyrus as a sparsification module.** A main cortical input to the hippocampus is routed through the dentate gyrus (DG) (Amaral et al., 2007). DG activity is characteristically sparse, with only about 2–5% of granule cells active in a given environment (Henze et al., 2000; Leutgeb & Leutgeb, 2007). We model the DG as a linear projection of visual features followed by batch normalization (running estimates of mean and variance, retention rate 0.95 per minibatch) and a high activation threshold to match the sparsity of activity ($\sim 2.5\%$).

**CA3 as a sequence-generating shift register.** We model CA3 as a linear RNN shift register that propagates inputs as theta sequences (Fig. 1C and E, CA3) and, for this paper, we keep it fixed to isolate the effects of intrinsic sequence generation from effects potentially induced by recurrent plasticity. The DG provides input $u_t \in \mathbb{R}^F$ over $F$ features. Each feature is assigned a dedicated prewired sequence of length $\ell$, so the CA3 state is $X_t \in \mathbb{R}^{F\ell}$. Motivated by hippocampal theta sequences (Dragoi & Buzsáki, 2006; Foster & Wilson, 2007; Leibold, 2020), we parameterize the total number of sequence units as $\ell = L + (R - 1)$, where $L$ is the number of theta cycles and $R$ the number of active units per cycle.

**Single-feature dynamics ($F = 1$).** Let $x_t \in \mathbb{R}^\ell$ denote the CA3 state for a single feature and $u_t \in \mathbb{R}$ the corresponding DG input. Sequence propagation is

$$x_{t+1} = S x_t + J u_t, \tag{1}$$

where $S \in \mathbb{R}^{\ell \times \ell}$ is the shift operator and $J \in \mathbb{R}^{\ell \times 1}$ injects the input into the first $R$ slots:

$$S = \begin{bmatrix} 0 & 0 & \dots & 0 \\ 1 & 0 & \dots & 0 \\ & \ddots & \ddots & \vdots \\ 0 & \dots & 1 & 0 \end{bmatrix}_{\ell \times \ell} , \qquad J = \big[\underbrace{1, 1, \dots, 1}_{R \text{ times}}, \underbrace{0, 0, \dots, 0}_{L-1 \text{ times}}\big]^T \in \mathbb{R}^{\ell \times 1}. \tag{2}$$

Thus a transient input $u_t$ creates activity in the first $R$ positions which is then shifted one step per timestep along the length-$\ell$ register.

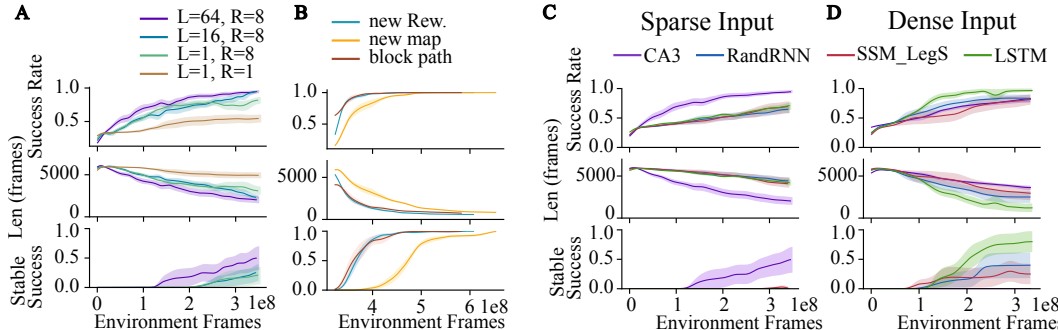

Figure 2: Training performance **A** Agents with different sequence length $L$ and repeat $R$. Line and shaded area are mean and s.e.m. across 6 random seeds. Success Rate: fraction of episodes where the goal is reached. Len: number of frames to reach the goal. Stable success: the rate of having 100 consecutive successful episodes. Metrics were Gaussian smoothed, $\sigma = 6 \times 10^6$ frames. **B** The best performing agent with $L = 64$ and $R = 8$ across all seeds was tested for its transfer learning. New Rew.: new reward location at the lower left corner. new map: a randomly generated map with the same statistics. block path: new walls are added to block paths to the reward, while the previous blocking walls were removed. **C** and **D** Performance of agents with different recurrent modules and inputs. **C**: Sparse input. **D**: Dense input where batch-normalization and high thresholding was removed. CA3: our CA3 model with $L = 64$ and $R = 8$. RandRNN: randomly initialized fixed RNN of the same state size. SSM_LegS: fixed SSM HiPPO-LegS from Gu et al. (2020) with the same state size. LSTM: trainable LSTM with matching number of parameters.

**Multiple features** ($F > 1$). Each feature dimension evolves independently under the same dynamics. Stacking all $F$ sequences with Kronecker expansion gives the block-structured update

$$A = I_F \otimes S, \qquad B = I_F \otimes J, \tag{3}$$

where $I_F$ is an identity matrix that has the size of the number of DG features. $S$ and $J$ are the same as the single feature case (For an extended explanation, see Appendix B.2). and the full CA3 dynamics

$$X_{t+1} = A X_t + B u_t, \tag{4}$$

with fixed recurrent matrix $A$ and input matrix $B$.

## 2.4 ACTOR CRITIC NETWORK

The activity of hippocampal units is fed into a decoder module with two linear/fully connected (FC) layers with ReLU activation. Actions and value are computed as the linear readout from the decoder module. We trained the agent with a standard advantage actor–critic objective (policy-gradient + value-baseline + entropy regularization) as implemented in Sample Factory (Petrenko et al., 2020; Espeholt et al., 2018).

## 3 RESULTS

### 3.1 BEHAVIORAL PERFORMANCE

Training the naive agent ($L = 64, R = 8$) on one fixed reward location in the maze exhibits robust performance measures after about 350 million frames (Fig. 2A). Thus our model effectively maintains information about sparse inputs in its trajectories of the RNN-like dynamics across time like in a reservoir (Jaeger & Haas, 2004). This view is further supported by our simulations with reduced sequence length showing inferior performance (Fig. 2A). In the most extreme case ($L = 1, R = 1$) without sequences, essentially a pure feedforward architecture corresponding to a brain with DG output bypassing CA3, the agent did not achieve robust behavior. While agents with reduced sequence lengths ($L = 1, 16, R = 8$) still can express reasonable success rates and trajectory lengths, their behavior is considerably more unstable as evidenced by the low fraction of consecutive successes (Fig. 2A). The agent's performance is stable within a wide range of $R$ and becomes more sensitive to

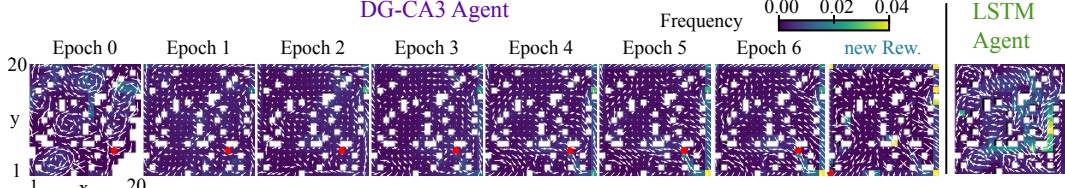

Figure 3: Evolution of occupancy over the course of learning. Color represents a normalized measure of the amount of time points the agent spent at a location. Mean head directions at a location were visualized in arrows. From epoch 5 onwards, the agent has a preference to reach the bottom right corner (independent of the random starting location) and proceed to the goal (red star) from there. The last panel shows the behavior of fully trained dense LSTM agent.

Table 1: Performance comparison across architectures. Training steps indicate the number of environment frames (in millions) required to reach 80% success . "✗" indicates that the threshold was not reached in all random seeds within 350M frames.

| Input | CA3 | Random RNN | HiPPO-LegS | HiPPO-LegT | HiPPO-LagT | LSTM |
|---|---|---|---|---|---|---|
| Sparse (Steps) | 173.6±77.6 | ✗ | ✗ | ✗ | ✗ | ✗ |
| Sparse (Succ.) | **0.86±0.10** | 0.51±0.12 | 0.52±0.11 | 0.57±0.15 | 0.65±0.10 | 0.56±0.06 |
| Dense (Steps) | ✗ | ✗ | ✗ | ✗ | ✗ | 135.9±27.6 |
| Dense (Succ.) | 0.71±0.07 | 0.78±0.15 | 0.64±0.21 | 0.65±0.28 | 0.38±0.02 | **0.93±0.09** |

$R$ when running speed is slower (Fig. A1 and Tab. A3). Transfer learning for new reward location and blocked paths, however, requires only about 50 million frames, and transfer learning to a new map requires about 150 million frames. These indicate that the agent was able to form a generalizing representation of the map and task (Fig. 2B).

The agent equipped with a sequence-generating module (CA3; eq.4) learns faster under sparse DG input than an agent in which CA3 is replaced by an LSTM core, resembling the SOTA architecture of Hessel et al. (2019) on the DMLab-30 benchmark (Beattie et al., 2016), from which our environment is adapted. Our CA3 module also outperforms state-space model HiPPO-LegS (Gu et al., 2020) and, as an additional control, randomly initialized RNNs, indicating that its theta-sequence dynamics provide a distinct advantage (Fig. 2C).

Crucially, this advantage is confined to the sparse-input regime. Under dense input, LSTMs perform better—consistent with prior reports (Hessel et al., 2019)—and our CA3 module performs worse than HiPPO-LegS and random RNNs (Fig. 2D). We also tested other variants from Gu et al. (2020) and they performed similar to HiPPO-LegS (Tab. 1, Fig. A4). These results highlight a specific synergy between sparse representations and intrinsic sequence generator.

## 3.2 OCCUPANCY

We divided the training into 6 epochs and evaluated the behavior and learned representations at these checkpoints. The agent develops a stable trajectory after 4 to 5 epochs. Between Epoch 4 and Epoch 5, the agent learned to get around the obstacle in the upper part of the right edge and started to spend more time at the lower right corner before reaching the reward site (Fig. 3). The agent appears to develop a strategy of visiting locations with salient input/landmarks and converging from different starting locations to the habituated paths, even after switching to new reward location. This is consistent with the typical strategies employed by animals in familiar enviroments (Gibson et al., 2013). In comparison, the LSTM agent under dense sensory input has fewer converging trajectories before reaching the reward (Fig. 3 and Fig. A11, likely due to the goal information being more readily available in the visual input, thereby implementing a strategy more related to visual search.

## 3.3 PLACE FIELD ANALYSIS

In order to understand how the agent executes a successful strategy and whether the representations in the hippocampus-inspired model would also align with the known physiology of place cells, we

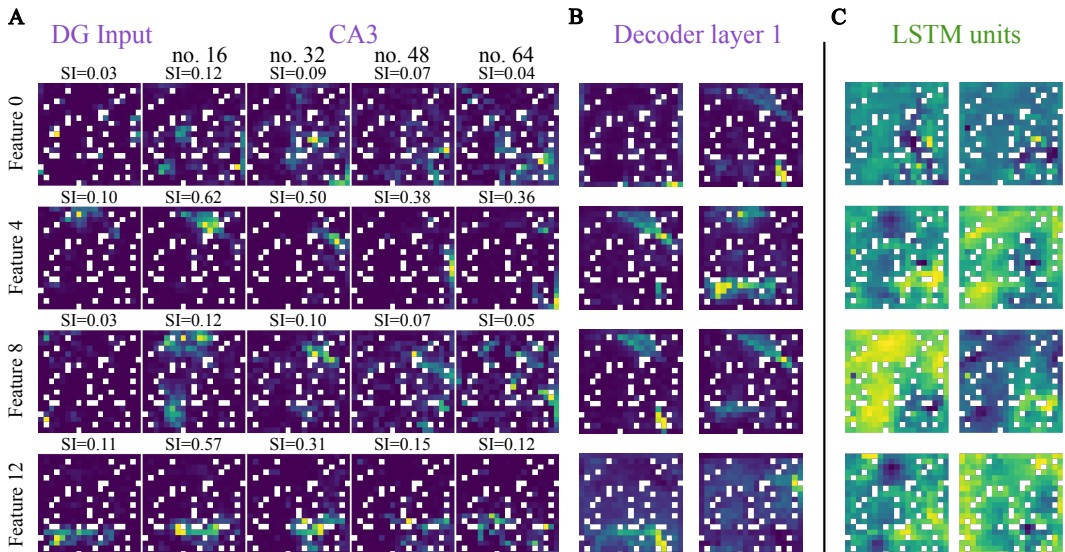

Figure 4: **A** Spatial tuning of DG and CA3 units from epoch 6. Pixel coordinates correspond to environment (black crosses correspond to walls). Each row shows the CA3 units ordered by their positions in the activity sequence. We selected 4 out of the 16 feature sequences for visualization. Spatial Information (SI): bits per time step. **B** Spatial tuning of randomly selected Decoder layer 1 units from epoch 6. **C** Spatial tuning of example LSTM output units in the dense LSTM agent. Colormaps are min-max scaled.

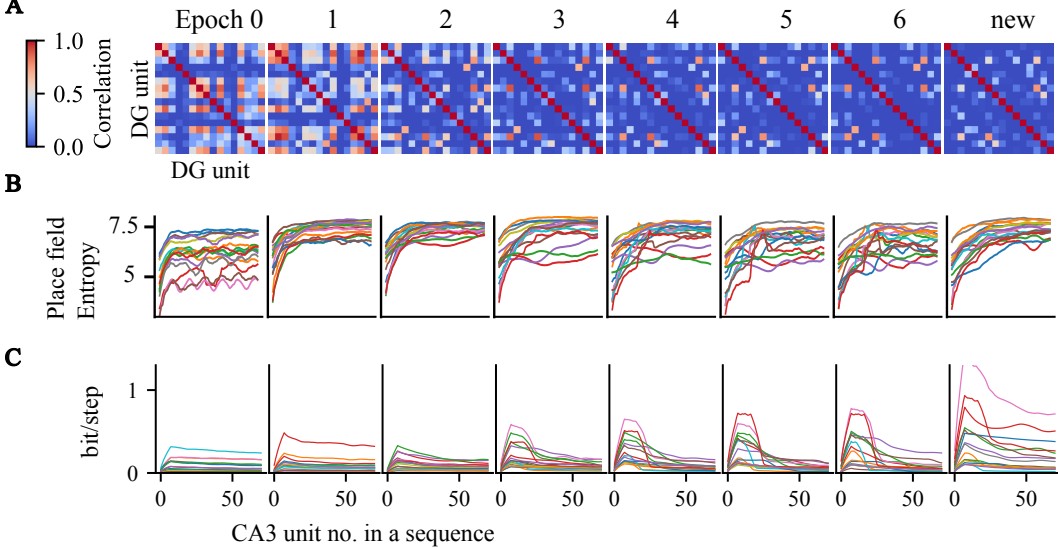

Figure 5: **A** Correlation (color code) of the 16 DG activity maps evoked by the visual features. Throughout training, the DG input to CA3 became increasingly orthogonal. **B** The spread of CA3 place field measured by entropy. The units are ordered by their positions in a sequence. Each sequence is indicated by a unique color across epochs. **C** Same as **B** but for the spatial information of CA3 units.

further computed activity maps of the units for the different layers of the model network. For the input units (DG), we (by construction) observe sparse activation that is distributed across few locations in the map that is further refined in the subsequent linear readout in decoder layer 1 (Fig. 4AB). Hidden units in the LSTM agent show non-localized tuning, distinct from typical place cells (Fig. 4C). More interestingly, learning the weights from the visual encoder to DG leads to orthogonalization, such that DG population activity over time develops a unique representation of individual locations (Fig. 5A). Units farther away from DG input in CA3 sequences show broader spatial tuning (Fig. 4A; quantified in Fig. 5B) matching previous reports (Jung & McNaughton, 1993; Parra-Barrero et al., 2021). The place fields of DG and CA3 units gradually stabilize through learning and exhibit remapping after adapting to a new reward location, measured by shifts of the place field center of mass (Fig. A10). In contrast, the input to LSTM agents did not show orthogonalization of the learned input projection (Fig. A14).

### 3.4 SPATIAL INFORMATION ANALYSIS

Qualitative inspection of the spatial tunings in Fig. 4 is also quantitatively supported by the distributions of spatial information (SI) of the neural activity maps: CA3 units activated $\sim 16$ steps after the input have larger SI (Fig. 5C). This corroborates our training results that agents with sequence length $L = 16$ are able to acquire the task reasonably well (Fig. 2). Units activated later in a sequence exhibit smaller spatial information (Fig. 5C).

Tracking the distribution of SI rates across training epochs, we observe a growing long tail indicative for the development of a neural space representation throughout all layers (Fig. A8). Furthermore, SI also increases with increasing hierarchical level of the network layer until the first layer of the decoder network.

To test whether SI is causal to behavior, we selectively permute the output weights of Decoder layer 1 units. When the permutation was done on the 32 units with the lowest SI, the performance was not impaired with respect to success rate and trajectory length (100% and 1065 frames). When the permutation was done on the 32 units with the highest SI, the success rate dropped by 4.9% and the average trajectory length increased from 1065 frame to 2794 frames.

### 3.5 POPULATION-LEVEL REPRESENTATIONS

We next examined population-level representations. We employ a novel method inspired by population vector (PV) correlation and representational (dis)similarity analysis (Kornblith et al., 2019; Kriegeskorte & Wei, 2021). It measures the mean population vector correlation in a pair of location bins grouped by the spatial displacement $\Delta x, \Delta y$. This measure can be interpreted as a spatial kernel learned by the network (Fig. 6). An unbiased representation of spatial geometry would show a kernel function that is isotropic, and smoothly and monotonically depends on the distance between locations bins. Conversely, the representation could be restricted to specific location bin pairs, displacement along specific orientation, specific spatial frequency, or simple visual resemblance. Kernels from all layers (DG, CA3, and Decoder layers) exhibit some dependence on distance throughout learning. However, CA3 kernels are smoother than the DG kernels. Both DG and CA3 show lower correlation values compared to the decoder layer. Notably, the Decoder layer 1 develops the most pronounced spatial tuning (Fig. 6A), consistent with the single-unit selectivity observed in Fig. A8. After the reward location was changed ("new"), kernels became less sharply defined, indicating that the representations are disrupted although behavior is adapted. Most layers in the agent with LSTM core and dense input did not show spatial kernels with strong displacement dependency. Only the LSTM output units showed a gradually refined spatial kernel during learning but it is strongly non-isotropic (Fig. 6B; Fig. A15). The dependence of representational similarity on distance and orientation is quantified in Fig. 6C.

Over training, a stable place code gradually emerges and remaps after learning the new reward location, indicated by the correlation of population vectors across training epochs (Fig. A9). Interestingly, the similarity between fully trained networks (epoch 6) and the novel reward condition ("new") is higher than between the naive (epoch 0) and trained (epoch 6) networks, suggesting that the agent acquires generalizable knowledge about the arena's spatial layout.

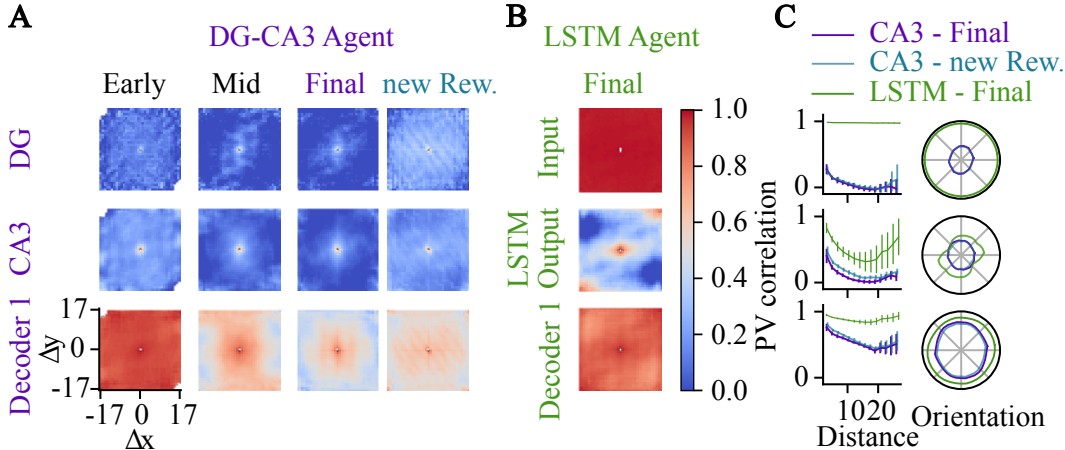

Figure 6: Kernel representation of distance. **A** Mean Pearson correlation (color code) of pairs of population activity vectors (PV) as a function of spatial displacement. Rows correspond to different layers (brain regions) of the network, columns to learning stages (epochs). **B** Same as **A**, but in dense LSTM agents. **C** PV correlation as a function of distance (left) and orientation (right).

## 4 DISCUSSION

In this study, we presented a minimal model of the hippocampus that enables navigation in a vision-based virtual environment and reproduces key phenomena of spatial representation observed in the mammalian hippocampus.

The cornerstone of our model — the sequential connectivity in CA3 — is biologically plausible and inspired by established findings on theta sequence firing (Dragoi & Buzsáki, 2006; Foster & Wilson, 2007; Yiu et al., 2022). By constraining CA3 recurrent connectivity as a dynamical reservoir and limiting training to input and output weights via reinforcement learning, we clearly isolate the effect of this theta sequence generator.

Agents equipped with the CA3 sequence module achieved superior navigation from sparse input (analogous to DG activity) compared to LSTMs of similar complexity or SSM-based cores. Crucially, this advantage was regime-dependent: under sparse input, CA3 dynamics and DG sparsification synergized to support robust navigation, whereas under dense input, conventional recurrent cores such as LSTMs performed better. Performance also degraded substantially when CA3 sequence length was ablated or shortened (Fig. 2), highlighting the functional role of theta sequence generator.

Analysis of neural activities revealed clear parallels with experimental observations of place cell properties, including robust place field formation (Fig. 4), population-level encoding of spatial distance (Fig. 6), progressive orthogonalization of DG outputs (Fig. 5A), and dynamic remapping triggered by changes in reward locations (Fig. A9 A10), consistent with experimental findings (Leutgeb & Leutgeb, 2007; Fenton, 2024).

**Interpretation.** The CA3 module expands sparse DG codes into a temporally smoothed canonical basis set, providing long-horizon history without the indiscriminate feature mixing of fully connected RNNs. This is especially beneficial under sparse input, where immediate sensory signals are limited and long-range context is critical for policy learning. The repetition parameter $R$ functions as a built-in prior of the temporal smoothness of latent states, thereby smoothing the CA3 spatial tuning via the agent's movement, at the cost of imprecise timing of input memory.

At first sight, sparsifying input to a recurrent core seems purely detrimental—leaving the agent "blind" most of the time. Yet it also filters out noisy, non-informative cues, making each supra-threshold input more reliably tuned to restricted regions of space. Through sequence propagation and policy learning, a spatially smooth representations can then be stabilized. This mechanism is consistent with habitual trajectories: neuronal activity late in a sequence only remains informative if the policy converges onto consistent paths.

Under dense input, LSTMs perform better, likely because the abundant weak signals can be accumulated to inform space, reducing the need for long-horizon buffering of strong signals. These intuitions are supported by comparative experiment where different levels of Gaussian noise were added to the pixel input to suppress weak signals. The DG+CA3 agent suffers less from increasing noise level than the LSTM agent (Fig. A5; Tab. A4).

Overall, these results suggest that different recurrent architectures are suited to different sensory regimes: sparse coding naturally complements sequential expansion, whereas dense input favors mixing-oriented recurrent dynamics. The agents' behaviors also diverged, resembling memory-driven navigation in our model versus more sensory-driven "visual search" navigation with LSTMs.

**Biological and computational relevance.** We proposed a parsimonious account of hippocampal theta sequences: they can be intrinsically maintained without requiring external input. While actual theta sequences in biological systems likely arise from multiple factors and vary across contexts (Chance, 2012; Schlesiger et al., 2015; Yiu & Leibold, 2023; Ahmadi et al., 2025), our abstraction nonetheless reproduced hallmark hippocampal phenomena, including robust place fields (Fig. 4), progressive DG orthogonalization (Fig. 5A), distance-dependent population kernels (Fig. 6), remapping after goal changes (Fig. A9 and Fig. A10), and, by construction, theta sequences (Foster & Wilson, 2007). These representational effects were linked to performance and align with evidence on sparse DG activity and CA3 sequences (Leutgeb & Leutgeb, 2007; Fenton, 2024).

Importantly, the visual encoder can be viewed as capturing the visually driven components of entorhinal cortex (EC) input to DG and CA3: the dorsal "where" and ventral "what" streams that project to medial and lateral EC (Wang et al., 2011). The depth sensor signal fed to Decoder can be seen as temporoamonic pathway from EC to CA1. To reflect the multi-modal nature of EC, additional modalities could be incorporated as input features.

Beyond explaining brain phenomena, our brain-inspired minimal model also proved useful as a module in competitive deep RL agents. The CA3 shift register can be viewed as a sparsely active reservoir that generates finite-length temporal bases, contrasting with the rotational modes of Legendre SSMs and the fading modes of Laguerre SSMs. Its structure resonates with recent shift–diagonal architectures (Fu et al., 2022), but tuned to sparse sensory regimes and navigation tasks. Thus, the model not only explains how theta sequences and place cells form but also provides a normative account of the computational effectiveness of this mechanism for navigation.

The comparison between our hippocampus-inspired model and an LSTM agent with dense input demonstrate the distinction of learned strategies and representations, even though the performance indicators are comparable (Figs. A15A14A11A12A13).

**Biological Predictions.** Our model predicts that larger environments or sparser inputs require longer sequences for successful navigation, consistent with developmental adaptability of sequence length (Wikenheiser & Redish, 2015; Farooq & Dragoi, 2019). More broadly, it suggests that hippocampal spatial representations could rely largely on intrinsic sequence-generating circuitry, with experience primarily shaping feedforward and readout connections. It offers an explanation for how place cells can persist despite lesions of entorhinal cortex (Hales et al., 2014; Schlesiger et al., 2015). The parsimonious nature of our model also provides a unified mechanism how navigation can be built upon sequences, no matter how they arise and how they differ across species. This is particularly important in species without prominent theta oscillations, e.g. bats show hippocampal sequences locked to wingbeats (Forli et al., 2025).

**Future Directions.** Several extensions follow naturally. On the biological side, incorporating local plasticity rules into the DG–CA3 pathway or CA3 readouts would align the model more closely with known mechanisms while preserving the benefits of prewired dynamics. Interactions with path integration in medial EC could further clarify their complementary roles in spatial cognition. Moreover, theta sequences have also been observed in other brain areas, raising the possibility of hierarchical coordination across regions.

These ideas resonate with developments in machine learning: structured dynamics akin to SSMs and linear attention are increasingly used in modern sequence models, including LLMs (Fu et al., 2022; Katharopoulos et al., 2020), suggesting that hippocampal-like motifs may illuminate principles of efficient long-range computation. The simplicity of our model also makes it interpretable: sequentially

connected CA3 units can be seen as structurally representing trajectories, which could in turn motivate algorithms that minimize internally measured trajectory length without relying on external reward.

On the ML side, combining sparse sequential reservoirs with learned SSMs may yield hybrid architectures capable of adapting flexibly across input regimes. More broadly, the bottom-up approach—focusing on circuit motifs rather than top-down expert informed objectives—suggests a path toward scaling to larger networks and more complex tasks.

**RL for neuroscience.** Our framework showcases how modern RL can serve as a testbed for computational neuroscience. By embedding a biologically inspired hippocampal circuit within an end-to-end reinforcement-learning agent, the model enables us to study brain computation in the context of the full perception–action loop, rather than in isolated modules for sensory processing, cognition, or motor control. Our approach thereby relates to ongoing discussions in machine learning regarding the "Bitter Lesson", which cautions against relying on handcrafted intermediate representations in favor of scalable end-to-end optimization. Importantly, our structural priors (sparseness, sequences) do not prescribe the representational solution; place fields, distance-dependent kernels, and remapping emerge through end-to-end learning. In this sense, physiological constraints act as an inductive bias that narrows the hypothesis space without sacrificing scalability, providing a middle ground between unconstrained black-box models and heavily engineered representational schemes. This perspective – imposing minimal structural constraints, such as sparse connectivity, rather than making apriori assumptions – is particularly relevant for association cortices such as hippocampus and prefrontal cortex, where emerging representational patterns are shaped jointly by sensory inputs, movement statistics, and behavioral goals (Lin et al., 2023).

An important conceptual point concerns how our representations relate to existing theoretical frameworks. Our CA3 representations relate to both successor-representation (SR) and reward/value-based frameworks, but with important distinctions. As in SRs, CA3 activity is policy dependent and shows a consistent temporal ordering: because the CA3 module propagates an intrinsic sequence, each unit predicts its downstream neighbors. This produces SR-like anticipatory structure, yet it arises solely from the fixed physiology-inspired architecture—CA3 does not learn a predictive map and receives no TD updates or discounted occupancy signal.

At the same time, unlike many previous RL studies of hippocampus (Kumar et al., 2022; Leibold, 2020), that takes allocentric input and analyze how policy shapes and utilizes them, our contribution is complementary: we show how localized, Gaussian-like place tuning can emerge from egocentric observations when combined with sparse DG input and intrinsic CA3 sequences. Our work also distinguishes itself from many other work that takes egocentric input but where the spatial similarity structure is already reflected in the input (e.g. small room where each wall has a different color; Vijayabaskaran & Cheng (2022); Raju et al. (2024)). Our environment is large maze, with uniform looking obstacles scattered in the arena, accounting for a more realistic discrepancy between visual representation and spatial representation.

The behavioral and representational divergence between the CA3-agent and the LSTM-agent illustrates a promising direction for future comparative work. Differences in qualitative strategies—not only performance—can reveal how architectural constraints shape navigation, representational geometry, and generalization, providing insights that are difficult to obtain from performance benchmarks alone.

**Conclusion** A minimal, sparsely driven sequence generator in actor-critic agent not only supports successful navigation but also gives rise to hippocampus-like spatial representations.

## 5 ACKNOWLEDGEMENTS

This work was funded by the German federal ministry for research, technology and space travel (BMFTR) under grant number 01GQ2510 and the BrainLinks-BrainTools Center.

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

APPENDIX

## A    LIMITATIONS

**Physiological Limitations**    Our model has fixed sequential connectivity in CA3, ignoring dynamic processes such as synaptic turnover, plasticity, and sequence length adaptations observed biologically. This simplification may limit the biological realism and generalizability of the model to adaptive neural processes. Additionally, the absence of direct entorhinal cortex inputs means we have not modeled potential interactions between hippocampal sequences and detailed sensory-motor driven activity.

**Conceptual Assumptions**    We assume sequential activity in CA3 arises exclusively from connectivity rather than sequential inputs, potentially oversimplifying hippocampal dynamics. This assumption excludes possible interactions between external sequential signals and intrinsic CA3 network dynamics, which could affect real-world predictive accuracy.

**Reinforcement Learning Methodological Limitations**    Our gradient-based reinforcement learning approach lacks clear biological analogs, as organisms likely utilize local Hebbian synaptic updates rather than global gradient propagation. Furthermore, the biological interpretation of the multilayer perceptron decoder remains unclear, posing challenges to direct biological interpretations of our results. Additionally, our use of asynchronous, batch-based training may not accurately reflect the temporal continuity or real-time single-organism learning dynamics seen biologically, thus potentially limiting ecological validity.

**Technical Limitations**    Our results depend on the environment specifications and hyperparameter settings. We chose a reasonably difficult environment setting where the difference between architectures can be highlighted. It does not imply the effect can generalize to all environments. We tested the agent with $L=64$ and $R=8$ on 5 other randomly generated maps, it could converge at a similar speed with the map shown in other results (Fig. A3). We also tested different frame-skip number, larger frame-skip leads to faster learning, and the effect of sequence length on performance is weaker (Fig. A16). This is probably due to the fact that our environment has sparse obstacles and that larger frame-skip leads to effectively longer sequence and memory. Most results in our report use 4 policies for population based training and 8 environments per worker. These parameters also change the dynamics of learning. having more policies made agents with $L=64$ learn slower and $L=8$ learn slower. Further systematic investigation of these factors is required to establish broader applicability.

Taken together, these limitations highlight important areas for further experimental and theoretical investigation.

## B    IMPLEMENTATION DETAILS

### B.1    ENVIRONMENT

| Description | Look (dx) | Pitch (dy) | Strafe | Forward | Fire | Jump | Crouch |
|---|---|---|---|---|---|---|---|
| Forward | 0 | 0 | 0 | 1 | 0 | 0 | 0 |
| Strafe Left | 0 | 0 | -1 | 0 | 0 | 0 | 0 |
| Strafe Right | 0 | 0 | 1 | 0 | 0 | 0 | 0 |
| Look Left + Forward | -20 | 0 | 0 | 1 | 0 | 0 | 0 |
| Look Right + Forward | 20 | 0 | 0 | 1 | 0 | 0 | 0 |

Table A1: Reduced action set used in DeepMind Lab experiments. Each action is defined over the 7-dimensional control space (`look, pitch, strafe, forward, fire, jump, crouch`).

Navigation task does not require complex actions, thus we reduced the action space used in IM-PALA (Espeholt et al., 2018), to facilitate training speed (Table A1).

Since we are mainly interested in modeling the way-finding aspect of navigation (Tolman, 1948; Kuipers, 2000), the MLP also receives average depth of the pixels, down-sampled to 10 horizontal pixels to aid motor control (going around walls and avoiding collisions). This procedure is supposed to mimic the sensory motor collision avoidance habits that do not depend on hippocampus. This depth information is directly routed to the decoder layer, so the representations in the recurrent core and its input are not directly influenced.

## B.2 MULTI-FEATURE DYNAMICS

As a more direct illustration for the recurrent dynamics of the sequence network $X_{t+1} = A\,X_t + B\,u_t$, we, here, provide the matrices $A$ and be for the multi-feature case:

$$
A = \begin{bmatrix} S & 0 & 0 & \dots & 0 \\ 0 & S & 0 & \dots & 0 \\ \vdots & & \ddots & & \vdots \\ 0 & \dots & & & S \end{bmatrix}_{F\ell \times F\ell} \quad , \quad B = \begin{bmatrix} J & 0 & 0 & \dots & 0 \\ 0 & J & 0 & \dots & 0 \\ \vdots & & \ddots & & \vdots \\ 0 & \dots & & & J \end{bmatrix}_{F\ell \times F} \quad . \tag{5}
$$

with

$$
S = \begin{bmatrix} 0 & 0 & \dots & 0 \\ 1 & 0 & \dots & 0 \\ & \ddots & \ddots & \vdots \\ 0 & \dots & 1 & 0 \end{bmatrix}_{\ell \times \ell} \quad , \quad J = \big[ \underbrace{1,1,\dots,1}_{R \text{ times}}, \underbrace{0,0,\dots,0}_{L-1 \text{ times}} \big]^T \in \mathbb{R}^{\ell \times 1}. \tag{6}
$$

Note that the rows in $A$ corresponding to the first time step of each feature only contain 0s, i.e., sequences can only be started from DG input and not from the dynamics itself.

## B.3 ARCHITECTURE

Table A2: Network architecture of the CA3 agent.

| Component | Details |
| --- | --- |
| **Encoder (ResNet)** | 3 conv blocks with residual layers, 16–32 channels, max pooling |
| **MLP head** | Linear (3456 → 256), ReLU |
| **DG projection** | Linear (256 → 16), BatchNorm (momentum 0.05, no affine), ReLU, intercept 2.43 |
| **CA3 (recurrent core)** | Fixed shift-register reservoir, size 16×(64+8-1)=1136 |
| **Decoder** | MLP: Linear (1136 + 10 → 128), ReLU, Linear (128 → 128), ReLU |
| **Critic** | Linear (128 → 1) |
| **Actor** | Linear (128 → 5) |

CA3 agent has learnable parameters in DG projection and from CA3 to Decoder, the CA3 output in the full version has 1136*128=149504 parameters. LSTM has learnable parameters $4*(m^2 + mn)$ where m is hidden size and n is input and output size. Solving this gives a hidden size of roughly 137, which we used in the implementation. SSM agents have matching state size with CA3 agent. They also have block-diagonal recurrent weights, where the weights in each block are obtained through the original implementation by Gu et al. (2020) with zero-order-hold discretizations.

## B.4 ACTOR CRITIC

We optimize an actor–critic objective with advantage estimates as in Espeholt et al. (2018), consisting of a clipped policy loss, a value regression loss, and an entropy bonus:

$$
\mathcal{L}(\theta, \phi) = -\mathbb{E}_t\big[\min\big(r_t(\theta)A_t,\ \text{clip}(r_t(\theta), 1-\epsilon, 1+\epsilon)A_t\big)\big] + c_v\,\mathbb{E}_t[(R_t - V_\phi(s_t))^2] - c_e\,\mathbb{E}_t[\mathcal{H}(\pi_\theta(\cdot \mid s_t))],
$$

where $r_t(\theta) = \pi_\theta(a_t \mid s_t)/\pi_{\theta_{\text{old}}}(a_t \mid s_t)$.

Here $R_t$ and the advantage estimates $A_t$ are computed using V-trace returns (Espeholt et al., 2018), which correct for off-policy updates arising in the asynchronous actor–learner setup of Sample Factory (Petrenko et al., 2020). This formulation—known as Asynchronous PPO (APPO)—combines PPO's clipped surrogate objective (Schulman et al., 2017) with IMPALA's V-trace corrections, enabling scalable training with many actors while maintaining stable policy updates.

### B.5 TRAINING CONFIGURATION

We trained agents using `sample-factory` (Petrenko et al., 2020) with the following setup.

**Environment.** frameskip 8, repeating action for 8 frames until getting the next observation. Each run used 32 workers $\times$ 8 envs/worker, with decorrelation up to 120s.

**Algorithm.** APPO (Espeholt et al., 2018), $\gamma = 0.99$, rollout length 64, recurrence 64, batch size 2048, 2 batches/epoch. Optimizer learning rate $2 \times 10^{-4}$.

**Architecture.** Visual encoder: pretrained ResNet on DMLab and the second layer of ResNet pretrained on ImageNet DG: batchnorm + ReLU, intercept 2.43, with 16 features. Decoder: 2 MLP layers of size 128.

**Population Based Training.** Enabled PBT with 4 policies, replacement gaps 0.05 (relative) and 0.2 (absolute), mutation start after 10M steps, period every 2M steps. Policy lag tolerance set to 35.

**Logging and checkpoints.** Training for 108k seconds, milestones every 5400s.

**Miscellaneous.** Seeds: [1111,2222,3333,4444,5555]. Device: CPU. Affinity pinning disabled. Inputs not normalized. Other parameters were default in sample-factory for DeepMind Lab.

### B.6 SKAGGS' SPATIAL INFORMATION MEASURE

We quantify how much information a unit's activity conveys about the agent's location in our discrete-time simulation by a "bits per step" version of Skaggs' spatial information (Skaggs et al., 1992). Let the environment be divided into $N$ spatial bins, and define:

- $p_i$ — fraction of timesteps (steps) spent in bin $i$ (occupancy probability),
- $\lambda_i$ — mean activity rate in bin $i$, measured in activity per step,
- $\lambda = \sum_{i=1}^{N} p_i \lambda_i$ — overall activity rate (activity per step).

First, the information conveyed per timestep is

$$I_{\text{step}} = \sum_{i=1}^{N} p_i \lambda_i \log_2\left(\frac{\lambda_i}{\lambda}\right) \quad \text{[bits/step]}.$$

This measures the average reduction in positional uncertainty (in bits) each simulation step provides.

Interpretation:

- Uniform firing ($\lambda_i = \lambda$ for all $i$) yields zero information ($I_{\text{step}} = I_{\text{spike}} = 0$).
- Elevated $\lambda_i$ in particular bins gives positive contributions proportional to $p_i \lambda_i$ (for bits/step) or $p_i (\lambda_i/\lambda)$ (for bits/spike).
- Bins visited rarely (small $p_i$) contribute less, guarding against over-weighing seldom-visited locations.

In our simulations, $\lambda_i$ and $\lambda$ are estimated from spatially binned sum of activity divided by the number of steps in each bin. Bins with $\lambda_i = 0$ are omitted (treating $0 \cdot \log 0 = 0$).

## C SUPPLEMENTARY TABLE

Table A3: The effect of parameter R at different running speed (frameskip). Performance is measured as the success rate at 150 million training frames. This is a summary of Fig. A1.

| Frameskip | R=1 | R=4 | R=8 | R=12 | R=16 | R=32 |
|---|---|---|---|---|---|---|
| 8 | 0.788±0.097 | 0.866±0.098 | 0.857±0.068 | 0.886±0.089 | 0.880±0.036 | 0.806±0.097 |
| 4 | 0.785±0.153 | 0.848±0.126 | 0.874±0.106 | 0.902±0.116 | 0.901±0.114 | 0.758±0.226 |

Table A4: The performance at different noise level. Performance is measured as the success rate at 100 million training frames. This is a summary of Fig. A5.

| Model \ Noise | 0 | 10 | 20 | 40 | 80 |
|---|---|---|---|---|---|
| DG+CA3 | $0.693 \pm 0.137$ | $0.492 \pm 0.167$ | $0.539 \pm 0.108$ | $0.685 \pm 0.186$ | $0.477 \pm 0.072$ |
| Dense LSTM | $0.638 \pm 0.063$ | $0.707 \pm 0.178$ | $0.495 \pm 0.010$ | $0.449 \pm 0.064$ | $0.373 \pm 0.008$ |

## D SUPPLEMENTARY FIGURES

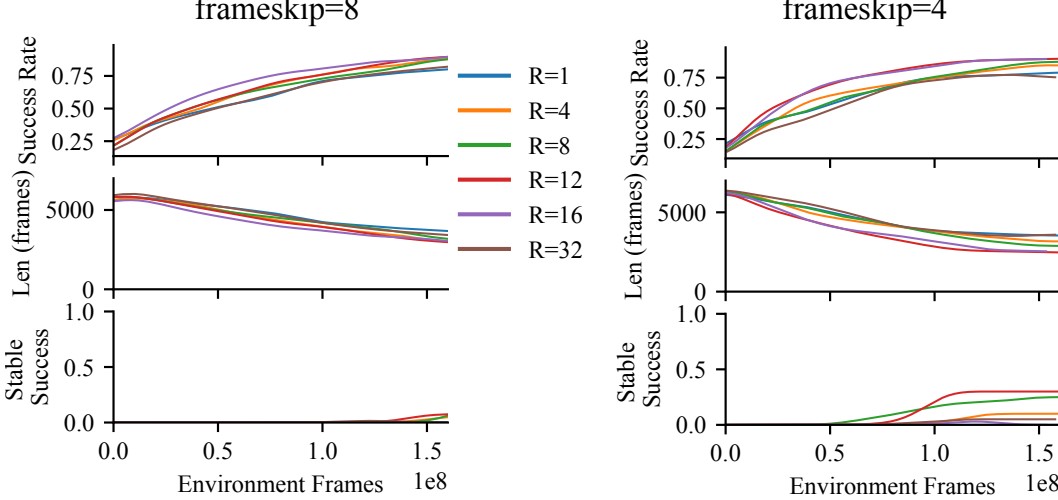

Figure A1: **Effect of parameter R with different running speed.** Training performance of CA3 agent with L=64 different R. Random seeds: [1111, 2222, 3333, 4444, 5555].

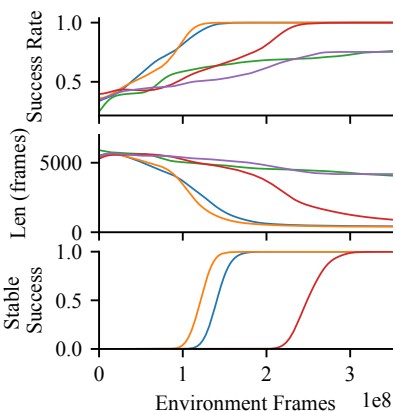

Figure A2: **Robustness of visual encoder.** Training performance of CA3 agent with the ResNet layer 2 output as visual encoder. Random seeds: [1111, 2222, 3333, 4444, 5555].

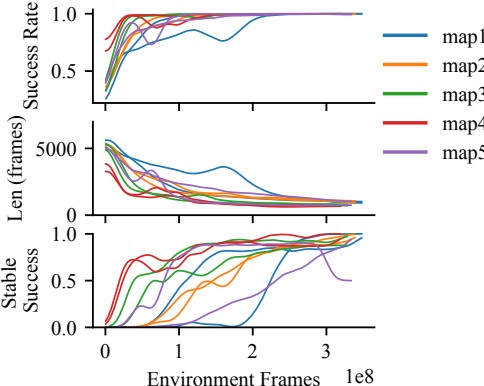

Figure A3: Training performance on 5 other randomly generated maps. Lines with the same color are results from random seeds [0, 42]

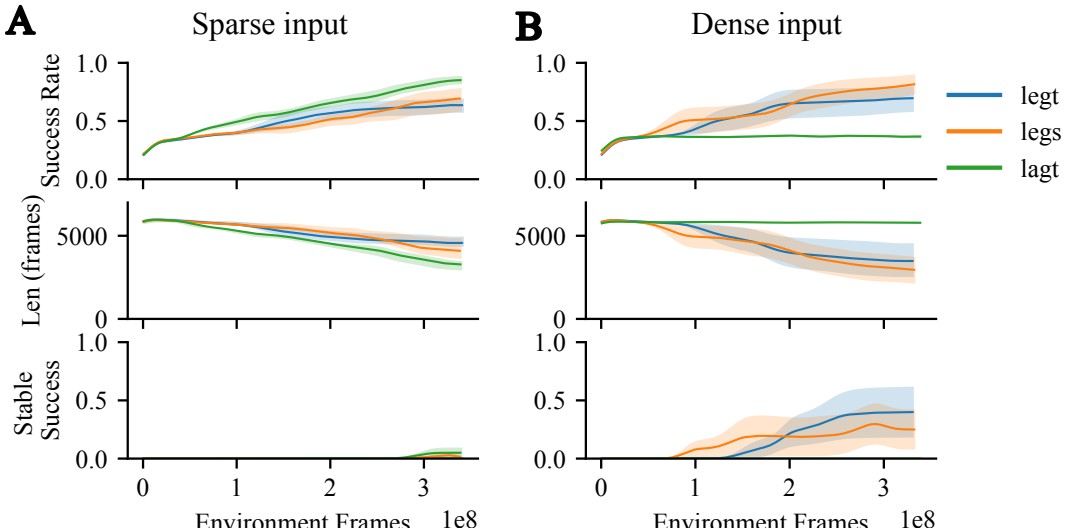

Figure A4: Training performance of agents with fixed SSM cores described by Gu et al. (2020). legt: Legendre bases with fixed memory horizon. legs: Legendre bases with infinite memory from the beginning of an episode. lagt: Laguerre bases, i.e. memory with decay. All three SSMs were implemented with the same state size as our CA3 module, each input feature is expanded into 64+8-1=71 states. The SSMs were fixed, with three additional learnable parameter controlling the input, recurrent scales and timestep length for discretizations, as a common practice described by Gu et al. (2021).

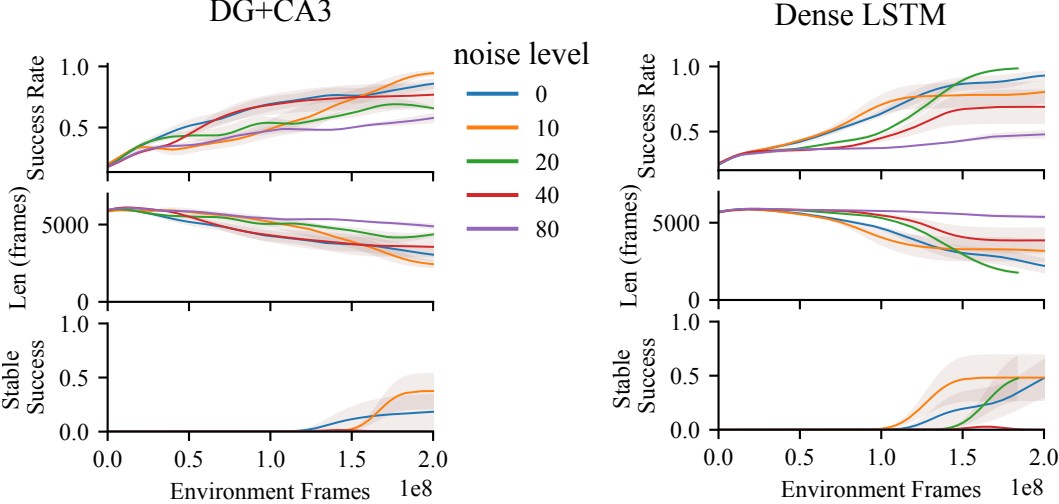

Figure A5: **Training performance of agents under different noise levels.** Independent Gaussian noise of different sigmas was added to the input image pixels (intensity 0-255). Random seeds: [1111, 2222, 3333, 4444, 5555].

[ 35  66  65  44  10  57  78 114  48  60  87  54  74 102  53 119]

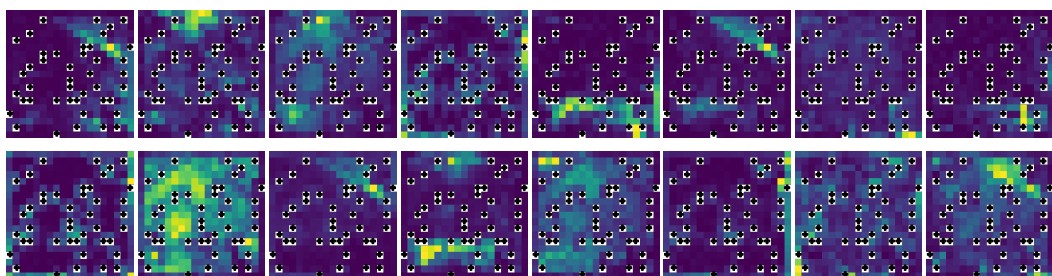

Figure A6: Rate maps from decoder layer 1 in the last training epoch. Showing the random 16 units out of the 50 units with the best SI. Unit id displayed on top.

[119  34  57  26 104  36   7 114 102  84  78  69 123  49  97  99]

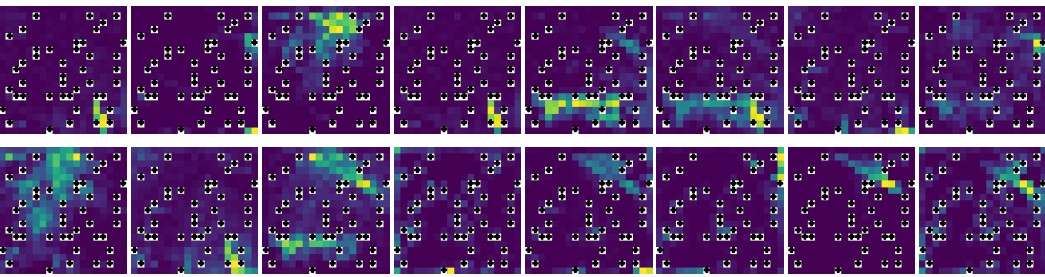

Figure A7: Rate maps from decoder layer 2 in the last training epoch. Showing the random 16 units out of the 50 units with the best SI. Unit id displayed on top.

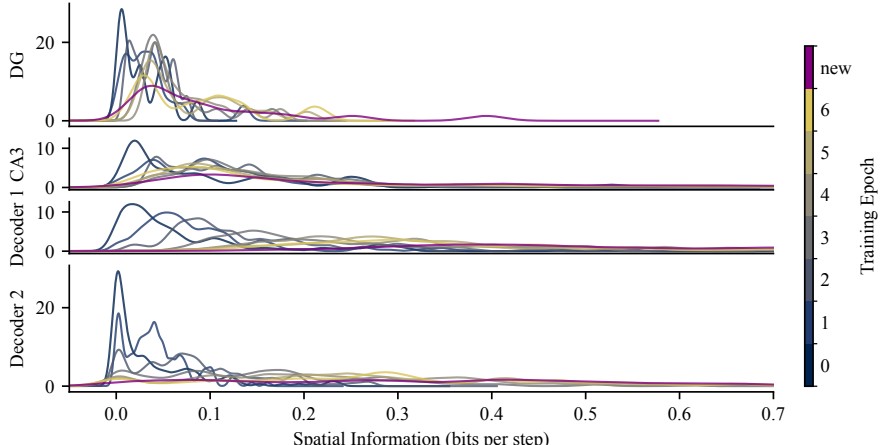

Figure A8: Distribution of SI rates for different layers (brain regions rows) over the progress of learning (colors) and change of reward site (purple). Y axes: unit fraction density.

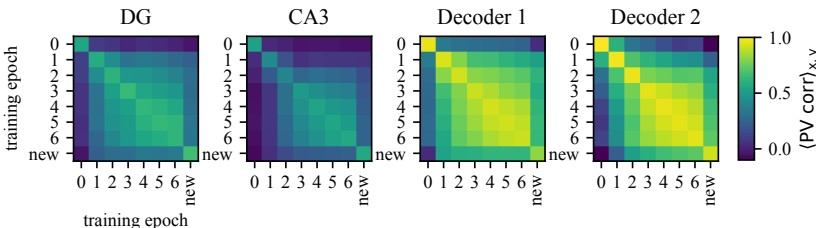

Figure A9: PV correlations across training. Although performance saturates by epochs 5–6, representational changes continue. The last epoch shows the effect of learning a new goal.

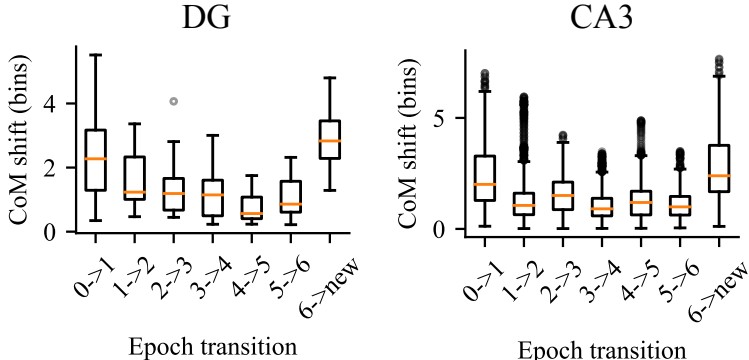

Figure A10: Convergence and Remapping. The center of mass (CoM) shift of place fields of individual units from epoch to epoch reveals higher stability in DG and remapping for a new goal.

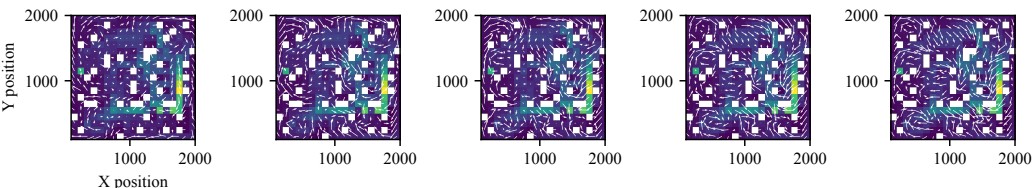

Figure A11: Occupancy map of LSTM agent's trajectories across training.

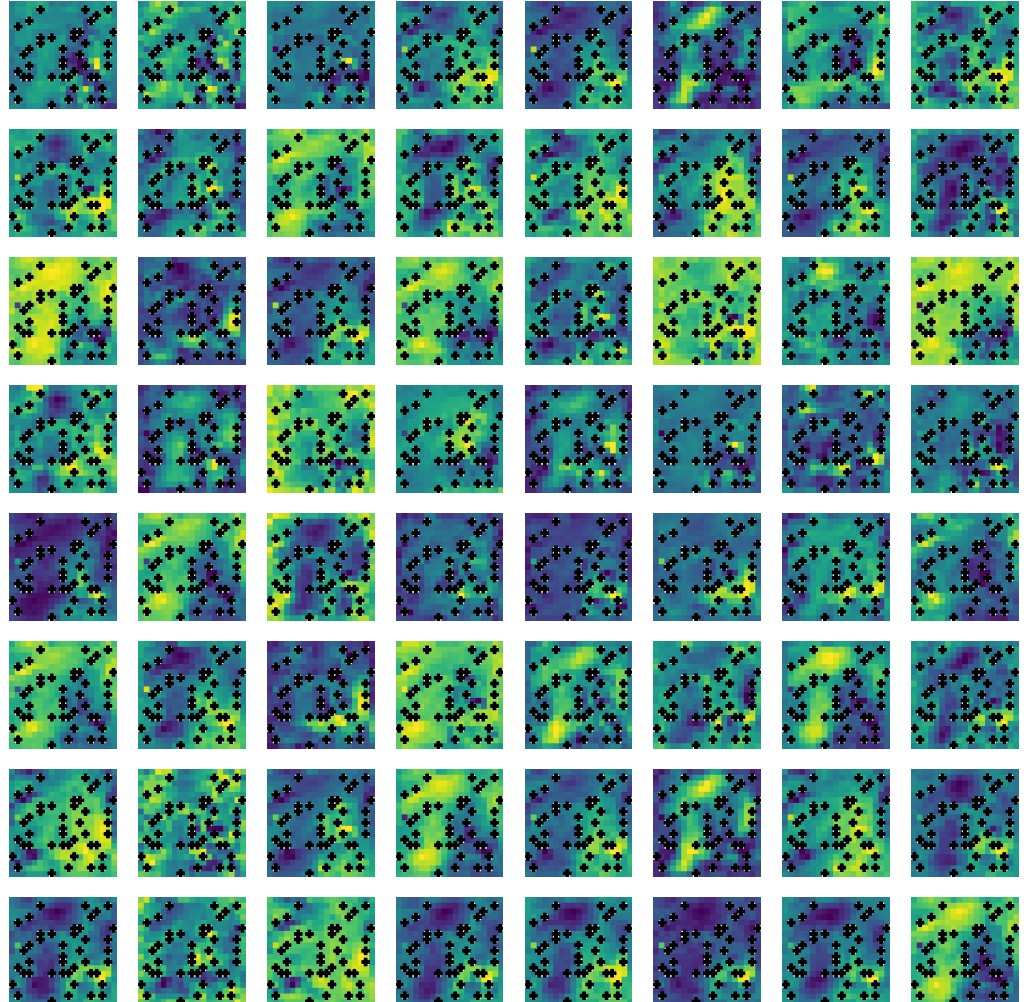

Figure A12: Place fields from the LSTM core (randomly selected units). Each panel shows the normalized spatial activity of one unit.

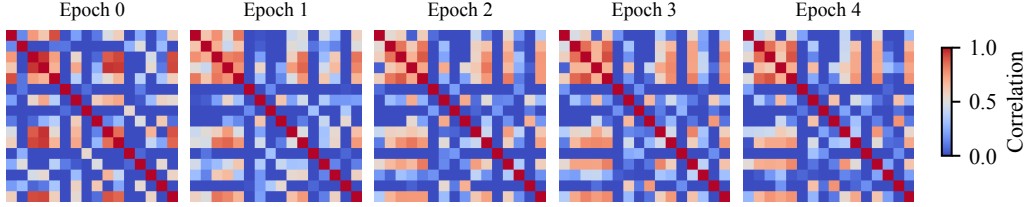

Figure A13: Population vector correlation across epochs. Color indicates Pearson correlation; rows show epochs 0–4.

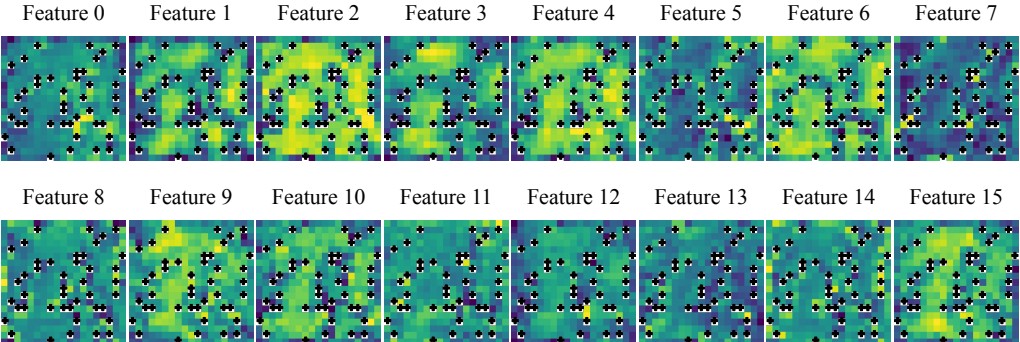

Figure A14: Learned dense input features in LSTM agent. Each tile shows the spatial receptive field of one feature.

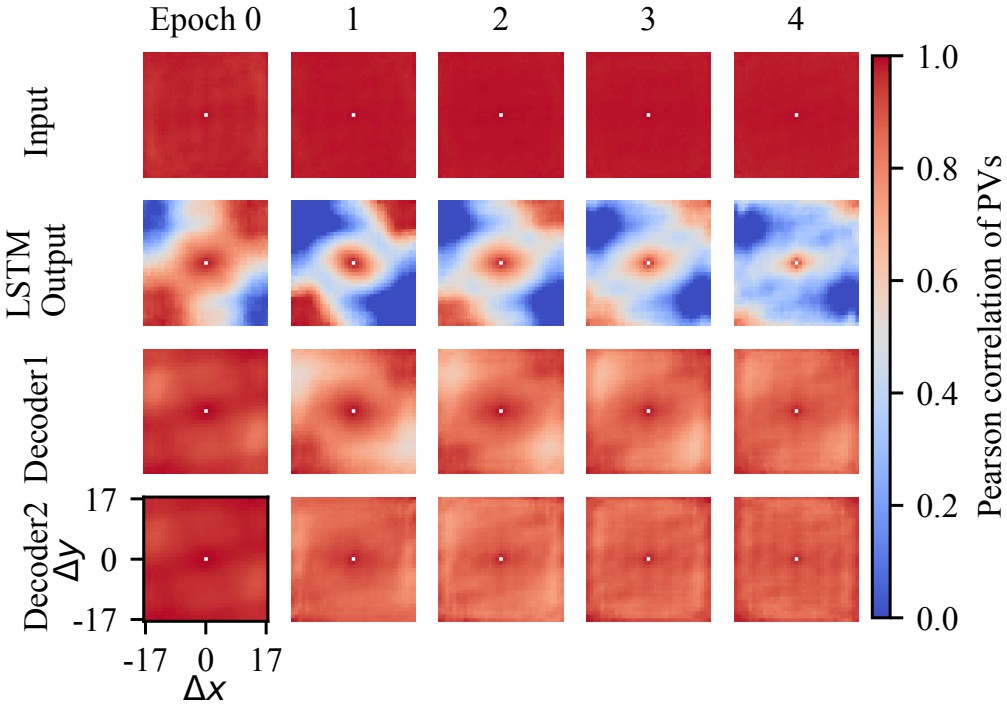

Figure A15: Displacement-dependent Pearson correlation kernel of population vectors. Axes show spatial shifts $(\Delta x, \Delta y)$; values are averaged correlations.

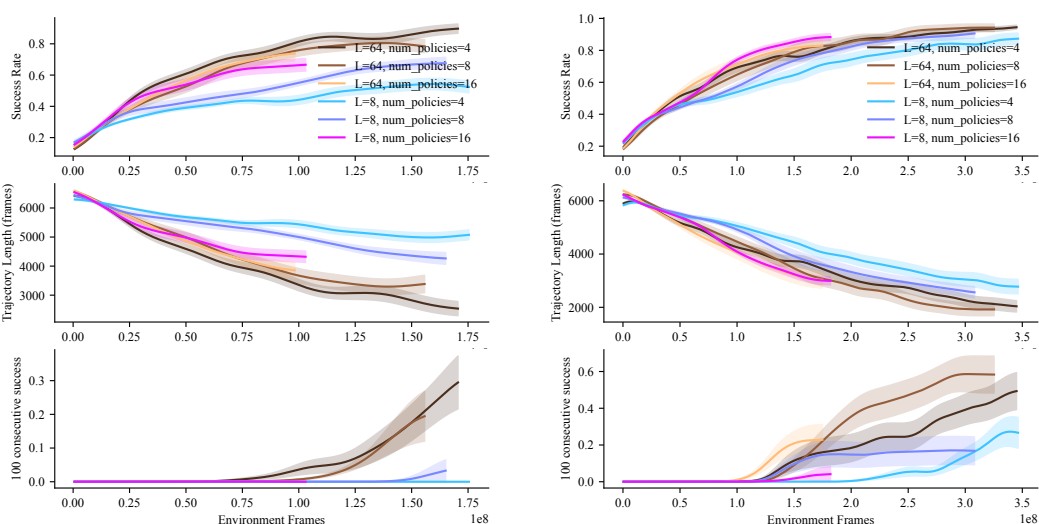

Figure A16: Training performance with different environment action repeats and number of policies in population based training. Left: environment frame skip / action repeat = 8. Right: environment frame skip / action repeat = 4. The frame skip controls the fine-graininess of actions, which is not required for the current navigation task. On the other hand, larger frame skip effectively make sequences propagate over longer traversals.

