# OpenReview forum: "Emergence of Spatial Representation in an Actor-Critic Agent with Hippocampus-Inspired Sequence Generator"
_ICLR.cc/2026/Conference — ICLR 2026 Poster_

### Official Review · Reviewer_EoBt · 2025-10-26

**Soundness:** 3
**Presentation:** 2
**Contribution:** 3
**Rating:** 6
**Confidence:** 4

**Summary:**

This submission studies the computational role of hippocampal sequences by constructing a neural network model with recurrence and examining how it performs on spatial navigation tasks. The authors find that sequence length is short, the model does not learn the task well (and struggles to generalize). Increasing the sequence length improves performance. The authors' model outperformed several other ML models when the inputs were sparse, but not dense. Investigation of the model revealed that place cell like responses emerged, as did the decorrelation of inputs.

**Strengths:**

1. Understanding hippocampal replay is an important open question and the authors tackle it with an interesting computational approach.

2. The results demonstrating the interplay between sequence length and sparsity are interesting and will be of broad interest to computational neuroscientists.

3. The authors have a detailed discussion section where they emphasize the potential impact their results could have on improving existing ML models. This will help broaden the impact of the work and help it fit more broadly in ICLR.

4. The experimental design, with obstacles and repeated visual inputs, was nicely done and makes the work more aligned with actual ethologically relevant environments.

5. The SI ablation study, where the weights were shuffled as opposed to just removed, was nice and convincingly shows spatial information is used.

**Weaknesses:**

1. The biggest weakness in my mind is that the explanation regarding the CA3 shift register was not so clear. In particular, I did not really understand: a) what it meant that each feature had a ``dedicated prewired sequence''; b) how the sequence was input to the decoder layers; c) what was learnable in the architecture. Later, in the Discussion, the authors mention that the sequence was a ``linear reservoir'', which I think clarifies some of these questions. But providing more details on this, in the section on the model, would be helpful. Perhaps including a figure demonstrating example sequences and a schematic highlighting how the sequence is passed to the next layer? There is also a minor typo in Eq. 2, where the number of 0s in J should be $L - R$, not $L - 1$.

2. The authors mention in the abstract, introduction, and discussion section the remapping between different environments. I never saw that results mentioned in the results (the figure being alluded to only in the discussion). If that's an important result, it should be highlighted earlier in the results section.

3. The writing in general felt terse. Not much detail was provided and many new paragraphs were used. This led to jumping between ideas/results and taking up space which then limited how much could be discussed. This was especially apparent in the Introduction, which did not feel very coherent or informative.

4. Bats and some birds do not exhibit theta oscillations. Despite this, these animals are extraordinary spatial navigators. I don't think this diminishes the authors' work, but I do think it is: a) worth commenting on; b) worth discussing how and why a lack of theta can be reconciled with their model/results.

**Questions:**

1. How exactly is the CA3 recurrent model being implemented and used by the rest of the model?

2. How do results showing a lack of theta oscillations in bats and birds line-up with the results the authors have shown?

---

> ### Author Response · Authors · 2025-11-22
>
> # Strengths
> We highly appreciate the supportive and insightful acknowledgments.
> # Weaknesses
> >The biggest weakness in my mind is that the explanation regarding the CA3 shift register was not so clear. In particular, I did not really understand: a) what it meant that each feature had a dedicated prewired sequence''; b) how the sequence was input to the decoder layers; c) what was learnable in the architecture. Later, in the Discussion, the authors mention that the sequence was a linear reservoir'', which I think clarifies some of these questions. But providing more details on this, in the section on the model, would be helpful. Perhaps including a figure demonstrating example sequences and a schematic highlighting how the sequence is passed to the next layer? There is also a minor typo in Eq. 2, where the number of 0s in J should be $L - R$, not $L - 1$.
>
> We thank the reviewer for the suggestions that would greatly improve the clarity.
>
> A dedicated prewired sequence means each sequence only propagates the input from one feature. It does not mix features.
> The sequences for all features were flattened and projected to the decoder layer 1 which  is a fully connected layer (with no additional structures imposed on the weights).
>
> The weight from visual features to DG, from CA3 to Decoder layer 1, from Decoder layer 1 to Decoder layer 2, and the linear readout for value and action are learned. The rest are fixed.
>
> We have included an updated illustration and caption in the new Figure 1 and extended the explanation for multi-feature case in the method and Appendix.
>
> We actually used R-1 number of 0s in our experiments (we stated there are L+R-1 neurons in each sequence).
>
> >The authors mention in the abstract, introduction, and discussion section the remapping between different environments. I never saw that results mentioned in the results (the figure being alluded to only in the discussion). If that's an important result, it should be highlighted earlier in the results section.
>
> We thank the reviewer for spotting this. The remapping was implied in the difference of SI and spatial kernel in epoch 6 and epoch new (new reward location). The direct quantification, population vector correlation (originally Fig. A7, now Fig. A9) was in the appendix for space reasons. We now followed the reviewer’s suggestion and mentioned it in the result section.
>
> We also included a new figure (Fig. A10) that quantifies the place field center-of-mass shift of individual units between epochs.
>
> >The writing in general felt terse. Not much detail was provided and many new paragraphs were used. This led to jumping between ideas/results and taking up space which then limited how much could be discussed. This was especially apparent in the Introduction, which did not feel very coherent or informative.
>
> We would like to thank the Reviewer for this suggestion. We have retouched the Introduction to improve reading flow, and would appreciate further feedback.
>
> >Bats and some birds do not exhibit theta oscillations. Despite this, these animals are extraordinary spatial navigators. I don't think this diminishes the authors' work, but I do think it is: a) worth commenting on; b) worth discussing how and why a lack of theta can be reconciled with their model/results.
>
> Theta oscillations are not clearly linked to place cell sequences in bats and birds. However there are analogous phenomena such as sequential activity that reflects ongoing movement. In bats, instead of being synced with theta rhythm, the place cell activations were found to be linked to 8Hz wingbeats ([Forli et al., 2025](https://www.nature.com/articles/s41586-025-09341-z)). More recently even prospective sweeps during flight were reported (SfN posters). The bird hippocampus, on the other hand, is still very little explored. It is still unclear whether there are strict homologies between avians and mammals on the circuit level. Much more basic research on birds is needed to be able to reasonably answer this question for birds.
>
> Our model is a parsimonious mechanistic model that could be applied to many different physiological implementations. It predicts the sequence-generating connections in CA3 and the sparse DG->CA3 activity are sufficient to produce decent navigation behavior and place tuning. Specifically, the propagation of sequences can be entrained to not just theta oscillation but any pace making mechanism such as wingbeats (could even be input-dependent or modulated by other inputs).
>
> We have included this in the Discussion section.
>
> # Questions
> >How exactly is the CA3 recurrent model being implemented and used by the rest of the model?
>
> We hope this question has been clarified by our response to weakness 1 above.
> >How do results showing a lack of theta oscillations in bats and birds line-up with the results the authors have shown?
>
> See response to weakness 4 above.

---

> > ### Comment · Reviewer_EoBt · 2025-11-22
> >
> > I thank the authors for their  clarifying responses. I better understand the work and I appreciate the added discussion on representations in brains without theta oscillations.
> >
> > I do not have any additional questions.

---

### Official Review · Reviewer_2sC9 · 2025-10-29

**Soundness:** 2
**Presentation:** 2
**Contribution:** 2
**Rating:** 2
**Confidence:** 5

**Summary:**

The authors introduce a biologically inspired model of the hippocampus focused on sequential connectivity from dentate gyrus (DG) to CA3, implemented within a reinforcement learning (RL) agent for visual navigation tasks. The model is designed with theta-sequence firing, while using sparse DG inputs. Navigation experiment demonstrates the model reproduces place field formation, remapping with reward changes, and orthogonalization of trajectories. The CA3 sequence generator helps the agent navigate more efficiently under sparse input conditions compared to standard RNN models.

**Strengths:**

- The architecture is constrained to reflect biological hippocampal anatomy, specifically the DG-to-CA3 connectivity and sparse input regimes.
- The model captures some experimental observations including place field formation, population-level spatial encoding, and remapping to new rewards.
- The baseline comparative analysis across different recurrent architectures suggests the functional significance of the sequence-generating module.

**Weaknesses:**

- Inconsistent performance: The CA3-based agent does not outperform LSTM or random RNNs in the dense input regime, and in some benchmarks, random RNNs do better than the CA3 module. I wonder about the generality and robustness of the sequence generation module. Is sequence generation necessarily a built in feature of the hippocampus or is it a consequence of something else? The current results do not support sequence generation in the hippocampus for faster learning.

- Ambiguous role of RL signals: The connection between reinforcement learning algorithms (e.g., A2C/TD error) and the emergence of hippocampal representations is not clearly explored. How does learning a policy or value function specifically influence place fields emergence and remapping (Kumar et al. 2025 ICML)? Could we get the same phenomena if we do behavior cloning or other supervised learning regimes? I do not see the point of using RL in this study other than to model navigation behavior.

- Insufficient clarity on planning: The authors do not analyze how the CA3 sequence generator supports planning, forward sweeps (Ito et al. 2015 Nature; Pfeiffer & Foster 2013 Nature), or predictive representations (Gardner et al. 2018). E.g. how do reward or environmental changes translate to different planning behaviors. Is your definition of a plan (rapid adaptation) different from policy learning (gradual learning)?

- Comparative analysis is limited: There is limited discussion on the contribution of sequence generation versus input thresholding in the observed performance with sparse inputs. Why does the proposed model only perform better with input thresholding and worse with dense inputs? More rigorous ablation studies might be relevant to disentangle these factors.

**Questions:**

1. Is it not expected that a CA3 module with high-thresholded, 2.5% sparsity will outperform other RNNs with sparse input (as in Fig. 2C)? Previous work (Kumar et al. 2022 Cerebral Cortex) has shown improved learning with higher thresholding but only to a point; how does your contribution differ?

2. Similarly, is it not trivial that different trajectories post-training will result in different CA3 sequences?

3. Could a chaotic attractor or standard reservoir network (e.g. Kumar et al. 2022 Cerebral Cortex) achieve similar benefits as your sequence generator?

6. To what extent do your representations align with the successor representation framework (Stachenfeld et al. 2017 Nature Neuro.), or are they more directly tied to reward/value learning signals (as in Kumar et al. 2025 ICML)?

References:
- Kumar et al. 2025 (https://icml.cc/virtual/2025/poster/46112)
- Ito et al. 2015 (https://www.nature.com/articles/nature14396)
- Pfeiffer & Foster 2013 (https://www.nature.com/articles/nature12112)
- Gardner et al. 2018 (https://royalsocietypublishing.org/doi/full/10.1098/rspb.2018.1645)
- Kumar et al. 2022 (https://doi.org/10.1093/cercor/bhab456)

---

> ### Author Response · Authors · 2025-11-22
>
> # Strength
> We thank the reviewer for acknowledging the biological grounding of our DG→CA3 architecture, its ability to capture key hippocampal phenomena, and the comparative analyses.
> # Weakness
> ## Inconsistent performance across regimes
> This concern assumes our goal is SOTA performance across all regimes. Our aims are (as stated in the ms):
> - to propose CA3 theta sequences arise intrinsically from recurrence rather than sequential input;
> - to demonstrate that this mechanism is specifically advantageous in the biologically motivated sparse–DG regime.
>
> In this regime, the CA3 agent performs strongly (Fig. 2C,D), matching dense-input SOTA levels despite much sparser (biologically realistic) activity.
>
> LSTMs perform well with dense input but do not produce hippocampal signatures (localized place fields, DG-like orthogonalization; Fig. A9–A10), making them poor hippocampal models even if performance is similar. CA3 requires informative sparse input to propagate sequences effectively, and the observed regime-dependence reflects the central mechanism of our study.
> ## Role of RL
> Using RL to model navigation behavior is essential. We test whether intrinsic CA3 sequences give rise to spatial representations from egocentric vision in the context of learning a navigation policy. CA3 feeds into the policy head; policy learning shapes DG responses, trajectory statistics, and thus CA3 place fields. This behavior–representation coupling would not be captured under behavior cloning or supervised learning, where trajectories are not influenced by representation. RL is also the ethologically natural setting for navigation.
> ## Clarity on planning
> Planning is not the research question of this work. Other studies interpret hippocampal sequences through the lense of planning, but in our model the sequence is a direct consequence of physiology, not necessarily related to introspective planning / counterfactual reasoning. Preplay could be explained by decoding the prewired chain. Ito et al. show planning-related activity is more CA1-dominated, consistent with our rigid CA3 dynamics.
> ## Why does sequence module need input thresholding
> CA3 propagates DG input through a fixed chain (a Jordan block with eigenvalue 0), preserving input timing but requiring informative input. Thresholding filters out noisy input at the cost of input quantity, which benefits CA3 but not standard RNNs that could process the input in their recurrence. Pixel-noise experiments show the comparative advantage of DG thresholding as noise increases, supporting this intuition. Plot is included in the appendix.
>
> Table: Success rate at 100 million training frames. Noise (sigma): gaussian noise added to the input pixels (intensity 0-255).
> |Model/Noise|0|10|20|40|80|
> |-|-|-|-|-|-|
> |DG+CA3|0.693±0.137|0.492±0.167|0.539±0.108|0.685±0.186|0.477±0.072|
> |Dense LSTM|0.638±0.063|0.707±0.178|0.495±0.010|0.449±0.064|0.373±0.008|
> # Questions
> ## Is CA3’s sparse-input advantage trivial (cf. Kumar et al. 2022)?
> Our contribution differs substantially:
> - We use only egocentric vision (no ground-truth spatial features), showing how Gaussian-like place fields emerge.
> - Kumar et al. do not compare against trainable RNNs such as LSTM, and do not model an explicit CA3 except assumed in the input.
> - Metric of Kumar et al. is memory capacity across many cue-reward associations; ours is navigation to a single reward, where sparsity impairs standard RNNs.
> - Our CA3 is a directional chain with eigenvalue 0, unlike Hopfield/reservoir architectures (eigenvalue ~1) in which sparsity is known to improve storage.
>
> Thus our results do not trivially follow from prior sparsification work.
>
> ## “Triviality’’ of trajectory-dependent CA3 activity
> Different trajectories following the same DG input produce the same CA3 unit sequence.
>
> They would lead to different place fields of these CA3 units. The key result is the bidirectional coupling, policy + DG → CA3 representation → policy, yields localized place fields and results in successful navigation instead of trivial solutions (circling, random policy, ignoring CA3).
> ## Reservoirs
> We evaluated SSMs (HiPPO), which are strong reservoirs (Mamba; Gu & Dao, 2024). Many reservoirs can exhibit sequential dynamics (revealed in the Schur/Jordan form). We chose a simple, controllable chain (a Jordan block with 0 eigenvalue) that allows transparent mechanistic interpretation and adjustable sequence length.
> ## Relation to other work
> Our CA3 exhibits SR-like properties (policy dependence, temporally ordered states) but does not compute discounted future occupancy or TD-based predictions. It also does not encode value directly; reward influences CA3 only indirectly through policy-shaped trajectories. Prior SR/value work assumes allocentric states; we focus on how localized place fields arise directly from egocentric vision. We have included this discussion in the revised manuscript. We welcome more specific questions from the reviewer.

---

### Official Review · Reviewer_vHh4 · 2025-10-31

**Soundness:** 3
**Presentation:** 2
**Contribution:** 3
**Rating:** 6
**Confidence:** 4

**Summary:**

This paper proposes a neuroscience-inspired architecture that combines sparse input - mimiking dentate gyrus (DG) - with a recurrent CA3 sequence generator to model spatial navigation. The authors demonstrate that under sparse input conditions, the CA3 architecture — implemented as a fixed shift-register that propagates theta sequences — outperforms LSTM cores of comparable size on an egocentric visual navigation task in a maze environment. The model develops place cell-like spatial representations and generalizes well to new reward locations, obstacles, and maps after training. Critically, the performance advantage disappears under dense input conditions, suggesting a specific synergy between sparse coding and sequence dynamics that may account for the emergence of hippocampal theta sequences.

**Strengths:**

**Neuroscience-AI integration:** The work effectively bridges computational neuroscience, AI, and reinforcement learning to account for the neuroscience-related problem of DG and CA3 synergy, theta sequences, and spatial representations emergence.

**Task validity:** The navigation task appropriately mirrors real-world neuroscience experimental paradigms, with the agent navigating using vision to locate rewards.

**Architectural interpretability:** The relatively simple architecture facilitates interpretation of individual components and minimizes confounding variables, allowing clear attribution of the effects observed.

**Generalization capacity:** The trained model demonstrates robust transfer learning capabilities when adapted to new reward locations, novel obstacle configurations, and new maps (Figure 2).

**Regime-dependent performance:** The demonstrated input-regime specificity provides evidence for the functional role of hippocampal sequences: CA3 outperforms LSTMs and alternative architectures under sparse input (~2.5% activity), while underperforming under dense input conditions. This specificity offers mechanistic insight into why theta sequences may have emerged under sparse input conditions.

**Weaknesses:**

**Incomplete parametric analysis:** The relationship between input density and CA3 performance (also w.r.t. LSTM or alternatives) requires systematic investigation. The paper tests R=1 only with L=1, leaving unexplored critical control conditions such as L=64 with R=4 (more sparsity) or L=64 with R=16 (less sparsity). A thorough parametric sweep varying R while holding L fixed would strengthen conclusions about the sparse-input CA3-DG synergy.

**Missing entorhinal cortex input:** The model considers only DG input to CA3, omitting entorhinal cortex projections entirely. The absence of discussion regarding this major hippocampal input limits biological completeness.

**Insufficient quantitative support for spatial tuning claims:** The claim of line 261 "Units positioned later in CA3 sequences (e.g. columns no.48 and no. 64 in Fig. 4) show broader spatial tuning as reported experimentally" lacks clear visual support in Figure 4 and requires quantitative validation.

**Unclear novelty and utility of spatial kernel measure:** Section 3.5's spatial kernel analysis does not clarify whether this measure is newly introduced or standard in spatial representation studies. The section primarily describes how the measure behaves across architectural modules and training epochs without drawing clear conclusions from these observations.

**(minor) Figure readability:** Font sizes in figures, particularly Figure 2, are too small for comfortable reading while going through the text.

**Questions:**

I believe the CA3 architecture (central to the paper) explanation of Section 2.3 is not entirely clear and I would have preferred an expansion in the appendix, expecially of the "Multiple Features" case. In my opinion, the meaning of "Each feature is assigned a dedicated prewired sequence of length l" (line 134) requires clarification, as does the definition of $I_F$, $S$, and $J$ in the multi-feature scenario (line 157).

---

> ### Author Response · Authors · 2025-11-22
>
> # Strength
>
> We would like to thank the reviewer very much for recognizing these contributions of our work.
>
> # Weaknesses
>
> ## Incomplete parametric analysis:
>
> We thank the reviewer for this point. Following the reviewer’s suggestion, we included a parameter sweep of R with L=64 under different running speeds (frameskip).
>
> Intuitively, R controls the trace length after a DG input event (the place field width). It could function as a built-in prior of the temporal smoothness of latent states (see Discussion), thereby smoothing the CA3 spatial tuning via the agent’s movement. On the other hand, the downstream Decoder weights could be adjusted to achieve similar effects, and overly large R would effectively destroy timing of input memory.
>
> As a result, the performance is stable within a large range of R. Using the original environment setting, R as small as 4 is enough to reach good performance. When the running speed is halved (effectively larger map or finer-grained timesteps), R=12 and 16 yield better performance than the others.
>
> We have included this result together with extended discussion in the revised manuscript (Fig. A1 and Tab. A3).
>
> **Table: Success rate at 150 million steps.**
> |Frameskip/R|R=1|R=4|R=8|R=12|R=16|R=32|
> |-|-|-|-|-|-|-|
> |8|0.789±0.097|0.866±0.098|0.857±0.068|0.886±0.089|0.880±0.036|0.806±0.097|
> |4|0.785±0.153|0.848±0.126|0.874±0.106|0.902±0.116|0.901±0.114|0.759±0.226|
>
> ## Missing entorhinal cortex input:
> We agree that entorhinal cortex (EC) inputs are a major pathway to the hippocampus. In our model, the upstream of DG is intended to reflect EC-derived transformations, and since the agent currently receives only visual observations, the medial and lateral EC can be viewed as the endpoints of dorsal “where” and ventral “what” visual streams (e.g., Wang et al., 2011 https://doi.org/10.1523/JNEUROSCI.3488-10.2011). Thus, while we do not explicitly model EC circuitry, our visual encoder plays the role of its visually driven components.
>
> We acknowledge that EC is inherently multimodal—integrating selfmotion (mEC grid cell system), olfaction (lEC), whisking (likely mEC; our depth sensory cue could be considered whisking input). Explicitly modeling all these pathways would require multiple domain-specific sensory models, which is beyond the scope of this work. Our aim is to focus on the computational contribution of the DG->CA3 sequence mechanism under minimal but biologically grounded sensory drive.
>
> Importantly, the framework is easily extensible: incorporating additional modalities corresponds to adding new input feature dimensions to the DG projection layer. Our current results show that egocentric vision alone (with limited depth cues projecting directly to decoder layer 1, which can be seen as EC->CA1 connection) is already sufficient for learning a cognitive map and achieving near–state-of-the-art navigation, supporting the relevance of our simplified setup.
>
> Following the reviewer’s advice, we have extended the discussion on EC in the revised manuscript.
>
> ## Insufficient quantitative support for spatial tuning claims:
> Following the reviewer’s suggestion, we have included a quantification of place field size, measured by the entropy of place field. It shows a clear increase along each sequence. We have included this in the main results (new Fig. 5B).
>
>
> ## Unclear novelty and utility of spatial kernel measure.
> The reviewer is correct that the exact same measure is not a standard choice. However, it is closely related to population vector correlations and representational (dis)similarity matrix (Kriegeskorte et al, 2008; Kornblith et al., 2019), our spatial kernel is a summary of the would-be (19x19)x(19x19) representational similarity matrix, grouping location bin pairs by their difference in x and y coordinates, regardless of the absolute x and y coordinates). This analysis views hippocampal spatial representation in the lens of metric learning. It examines if the similarity of representation reflects the physical distance between locations. Conversely, it could be biased to reflect only the difference in one direction (e.g. the kernel of LSTM agent primarily differentiates the upper-left and lower-right displacement) or reflect more of the similarity structure of the visual input (e.g. two obstacles could look the same even if they are far apart).
>
> We have extended the explanation of kernel measure in the Results.
>
> references:
>
> Kriegeskorte et al., 2008 https://www.frontiersin.org/journals/systems-neuroscience/articles/10.3389/neuro.06.004.2008/full
> Kornblith et al., 2019
> https://proceedings.mlr.press/v97/kornblith19a.html
>
> ## (minor) Figure readability
> Thanks for the suggestion. We have fixed figure 2 and will make sure the font sizes are more reader-friendly for the camera-ready version.
>
> # Questions:
> ## CA3 architecture explanation
> We have included an updated illustration in the new Figure 1E and extended the explanation for multi-feature case in the method and Appendix.

---

### Author Response · Authors · 2025-12-03
**Clarification of Scope**

We would like to briefly clarify the intended scope of the submission to help contextualize the reviews and rebuttal.

Our paper evaluates a hippocampal model by embedding it within the end-to-end training of a deep RL agent performing egocentric visual navigation. This framework allows us to (i) use realistically complex tasks, (ii) examine representations that are not simply input-driven but coupled with action generation, and (iii) preserve interpretability of the target brain region without oversimplifying its upstream or downstream signals. Within this setting, we find that intrinsic CA3-like sequential dynamics, when driven by sparse DG-like input, give rise to hippocampal spatial representations directly from raw vision. The central focus is the mechanistic interaction between sparse input and sequence-generating dynamics, providing a normative account of hippocampal physiology and a useful inductive bias for machine learning.

Reviewers vHh4 and EoBt evaluated the work within this scope and highlighted the biological grounding, emergence of spatial representations, regime dependence, control simulations, and clarity of the mechanistic contribution. Reviewer 2sC9 similarly noted as strengths the biological realism, alignment with experimental observations, and comparative analyses illustrating the functional significance of the sequence-generating module.

Some points raised by Reviewer 2sC9 concern broader topics—dense-input SOTA performance, planning-related interpretations, behavior-cloning or supervised alternatives, and successor-representation/value-learning comparisons—topics that extend beyond the problem our paper is designed to investigate. In the rebuttal, we clarified how these topics relate to, but differ from the research questions in our paper, and why they constitute follow-up directions rather than contradictions of our claims. We also explained the difference between prior work (such as Kumar et al.), which relies on place-field–like state inputs and evaluates memory capacity, and our setting, where place fields emerge from pixels during navigation in a much larger, feature-poor environment (uniform blue floor, yellow walls) that provides no unique visual cues for position—an important difference from the settings typically used in comparable studies. Since the mentioned previous work addresses a different scope with different evaluation criteria, not targeting those objectives reflects a difference in focus rather than a weakness of the submission. Likewise, our findings are not direct extensions of that work, as the assumptions, inputs, and task structures differ substantially.

We hope this clarification helps frame the scope under which the submission is meant to be evaluated, alongside the rebuttal and discussion. We appreciate the AC’s time and consideration.

---

### Meta-Review · Area_Chair_N1G2 · 2025-12-18

**Summary:**

The reviewers generally appreciated the neuroscience-inspired design of the model but found the technical description of the central CA3 sequence generator to be vague such that it would be difficult to reproduce. Specifically, the mechanisms behind the fixed shift-register and the assignment of prewired sequences to features were unclear, leaving the reviewers unsure about what aspects of the architecture were learnable versus fixed. There were also significant concerns regarding the model's robustness, as it failed to outperform standard LSTMs or random RNNs under dense input conditions, which raised doubts about the general utility of the proposed sequence generation mechanism outside of specific sparse regimes.

The reviewers also requested more rigorous hyperparameter analyses to disentangle the effects of sparsity and sequence length, noting that the results did not fully justify the specific synergy claimed between sparse coding and sequence dynamics. Furthermore, the biological completeness was questioned due to the omission of entorhinal cortex inputs and the lack of discussion regarding species that navigate efficiently without theta oscillations, such as bats. Finally, reviewers noted that key results regarding spatial remapping and tuning were either claimed without sufficient quantitative evidence or misplaced within the manuscript's structure, appearing in the discussion rather than the results.

**Reviewer Concerns:**

The authors provided a fuller explanation of the CA3 sequence generator and assignment of prewired sequences. They also performed additional hyperparameter investigations and gave further discussion on some of the pertinent issues raised (e.g. re theta cycles).

They did not fully address the concern that the model doesn't perform as well as LSTMs in dense input conditions, but they did provide a reasonable explanation for why this was outside the purposes of their paper.

**Reviewer Scores:**

The initial scores were 6,2,6. My guess is that one or both of the 6s would have increased their score. The 2 rating seemed overly-harsh given the review, and that reviewer may have been attempting to derail a paper that they considered a competitor to their own work. Thus, I'm not sure they would have raised their score, particularly since the authors largely side-stepped their critique. So, my guess is the final scores would have been 7,2,7. But, I think the 2 is not really a fair score for this paper, particularly given the comments from the other reviewers.

---

### Decision · Program_Chairs · 2026-01-26

Accept (Poster)